# Microbeam X-ray diffraction study of lipid structure in stratum corneum of human skin

**Naoto Yagi**[ID]*, **Koki Aoyama, Noboru Ohta**

Japan Synchrotron Radiation Research Institute, SPring-8, Sayo, Hyogo, Japan

\* yagi@spring8.or.jp

**Data Availability Statement:** All relevant data are within the manuscript and its Supporting Information files.

**Funding:** This work was supported by JST CREST Grant Number JPMJCR1521, Japan.

## Abstract

Human skin, not previously frozen, was studied by small-angle X-ray diffraction. The samples were folded so that a 6μm X-ray beam passed through the top layer of skin, stratum corneum. Diffraction patterns recorded with this method consisted of peaks at about $q = 0.5$, 1.0 and 1.4 nm$^{-1}$ in the direction perpendicular to the skin surface more clearly than in previous studies. These peaks are interpreted to arise from lipids between corneocytes. A simple unit of a linear electron density profile with three minima was used to account for the observed intensity profiles. Combinations of calculated diffraction from models with one, two and three units accounted for the major part of the observed diffraction pattern, showing the diversity in the structure of the intercellular lipids.

## Introduction

The top layer of human skin, stratum corneum (SC), serves as a barrier which protects human body from dehydration and penetration of unwanted substances. Its property is also important in tarnsepidermal drug delivery. SC comprises of inactive cells (corneocytes) in which keratin filaments are densely packed [1] and lipids such as ceramide, cholesterol, fatty acids and glycerides fill the space between the cells. The barrier function is considered to mostly depend on the lipids that fill the gap of less than 50 nm wide. Several lipid molecules, such as ceramides and sphingolipids, are arranged in a lamellar structure with their long axes lying across the gap. Many studies on the intercellular lipid structure have been carried out using electron microscopy, electron diffraction and X-ray diffraction. In most experiments, SC was chemically treated to prepare specimens suitable for each technique. In X-ray diffraction experiments, SC samples were often separated from skin by trypsin digestion [2–5] and investigated as whole SC. This method allows control of hydration and temperature and helps identify origins of diffraction peaks [2, 4, 5]. Studies on human [3, 6, 7], mouse [4, 5], and pig [2] SCs as well as human skin equivalents [8, 9] showed the presence of common basic lipid structure. X-ray diffraction studies on lamellae of lipid constituents of skin have been also made to investigate lipid phases [10–12]. For electron microscopy, it is common to go through the procedures of chemical fixation, dehydration and staining of samples before cutting thin sections [13]. Thus, an observed pattern of density in electron micrographs may not represent electron density in a natural specimen. So far, high-resolution structural studies under a condition that is

**Competing interests:** The authors have declared that no competing interests exist.

most similar to a living body were made using cryo-electron microscopy [8, 14]. For this technique, human SC is rapidly frozen by liquid nitrogen and then cryo-sectioned and observed by an electron microscope in a frozen state. Thus, it is possible to obtain more direct information on electron density distribution than conventional electron microscopy. However, to obtain structural information at higher spatial resolution, it is necessary to study SC under physiological conditions with X-ray diffraction. For this purpose, we performed small-angle X-ray diffraction (SAXD) measurements of SC on intact human skin.

Recently, there has been a report of an X-ray diffraction study on intact human skin that was cut into a fine (0.1 mm width) strip [6]. Using an X-ray beam with a diameter of a few micrometers, it is possible to investigate the depth dependence of lipid structures in SC. Our sample is a similar one (from the same supplier) but we employed a technique that does not require cutting a skin sample into a fine strip which might distort the structure. As was in a previous experiment on skin of a transgenic mouse [15], we folded a skin sample and passed an X-ray microbeam at the edge of the sample to obtain diffraction from SC so that we can study skin with X-rays under conditions as physiological as possible. The diffraction patterns thus recorded were found suitable for a detailed structure analysis for which we employed a simple tri-lamellar model.

## Materials and methods

### X-ray diffraction techniques

The SAXD measurements were performed at BL40XU (High Flux Beamline) in the SPring-8 synchrotron radiation facility (Hyogo, Japan). A quasi-monochromatic high-flux X-ray beam from a helical undulator ($\lambda$ = 0.083 nm) was focused with two mirrors laid horizontally and vertically [16]. In the experimental hutch, an X-ray beam of about 6 μm in diameter was obtained behind a collimating pinhole (5 μm in diameter) and a guard pinhole (100 μm in diameter) [17]. The X-ray flux was about $1 \times 10^{11}$ photons/s. The sample-to-detector distance was 1503 mm. The scattering vector $q = (4\pi/\lambda)\sin(2\theta/2)$ was calibrated with powder diffraction from silver behenate, where $2\theta$ is the scattering angle. X-ray diffraction patterns were recorded with an exposure time of 0.1 s using an X-ray image intensifier (V5445P, Hamamatsu Photonics, Hamamatsu, Japan) coupled to a CCD camera (C4880-50-24A, Hamamatsu Photonics) with 1024 x 1024 pixels [18]. The experiment was made at an ambient temperature of about 27 ˚C.

### Human skin samples

Six full thickness human skin samples without adipose tissues were purchased from Biopredic International (France). They were obtained from abdomen of healthy Caucasian females (36–56 years old) in cosmetic surgery eight days before the experiment, punched in a disc with a 10 mm diameter and immersed in a preservation medium. The skin samples were provided without any personal information on the donors except sex, ethnic origin, anatomical site, age, and BMI. As for the ethics concerning the samples and compliance with the laws, the supplier, Biopredic international, provides information (https://www.biopredic.com/rubrique-ethics)). From the nature of the surgery, the range of donors is limited. The samples were transported in a fresh state at 4 ˚C without freezing. Some of the skins were cleaned with chlorhexidine alcohol or betadine before surgery.

Just before the experiment, a skin sample was taken out of the preservation solution and blotted on a filter paper to remove excess solution. Then, it was folded onto a folded aluminum tape so that it formed a pointed edge at the top (S2b Fig). The ends of the sample were fixed together with a paper clip. The skin was mounted on vertical and horizontal translational

stages in air and an X-ray microbeam was passed through the SC at the pointed tip. The skin was moved vertically by 5 μm after each exposure to scan the entire thickness of the SC. This vertical scan was performed at five to ten different locations on each skin sample. The total measurement took 15 to 20 minutes. The skin sample was found moist after the experiment and change in the diffraction pattern due to drying was not observed throughout the experiment. A second scan at the same position provided identical results showing absence of a serious radiation damage, but the sample was shifted by about the diameter of the X-ray beam to avoid irradiating the same spot twice.

## Data analysis

Lamellar diffraction appeared perpendicular to the skin surface (Fig 1). A detailed consideration on the nature of the diffraction is described in S1 Appendix. A radial distribution profile of diffraction intensity was obtained by summing intensity along the length of an arch at each radial position. For presentation, intensity was multiplied by $q$ for the Lorenz correction. This is necessary to compare with intensity profiles obtained by simulations.

## Results

### Experimental

Fig 1 shows a galley of SAXD patterns from human skin. Broad diffraction peaks due to oriented lamellar-like structure are clearly visible. Fig 2 shows intensity profiles along the direction perpendicular to the skin surface (this direction is defined as the meridian) integrated along the arcs. Profiles at six successive positions separated by 5 μm are plotted with displacements. Data from the six samples are shown. The profile at the bottom (black) is where the peaks were first observed and that at the top was recorded after the beam moved by 25 μm into the skin sample. Including the beam size, total of about 30 μm of thickness was investigated. The main features are the three major peaks at about $q$ = 0.5, 1.0 and 1.4 nm$^{-1}$ along the meridian.

The diffraction profiles in Fig 2a–2e show three major peaks at $q$ = 0.51, 1.03 and 1.4 nm$^{-1}$, the first two of which may be the first and second orders, but their intensities do not correlate well at different depths: in most of the typical results obtained in this study, the peaks at $q$ = 1.03 and 1.4 nm$^{-1}$ are much stronger than that at $q$ = 0.51 nm$^{-1}$ in the region close to the skin surface (black and blue curves), while the 0.51 nm$^{-1}$ peak becomes stronger in deeper regions (red and brown curves). When measured at the depth where the peaks are most prominent, typically at 10 μm from where the peaks were first observed, the first peak was at $q$ = 0.511 ± 0.005 (mean ± standard deviation, range 0.504 to 0.515) nm$^{-1}$ and the second was at $q$ = 1.034 ± 0.013 nm$^{-1}$ (range 1.019 to 1.055). These values are averages over values from five skin samples, each of which was an average of values obtained at 5 to 10 different locations on the skin. The ratio of the two values is close to 2, suggesting that these are the first and second orders of the 12.3 nm periodicity (periodicity d is obtained by d = $2\pi/q$). Additionally, there is a broad peak at around $q$ = 2 nm$^{-1}$ which may be the 4th order of this periodicity. Fig 2 clearly shows that the peak at $q$ = 1.4 nm$^{-1}$, which was observed as a shoulder in a previous report on human skin [6], is indeed a separate peak as observed by Schreiner et al. [7].

When the X-ray beam is close to the skin surface, the peak at $q$ = 0.5 nm$^{-1}$ is usually weaker than that at $q$ = 1.0 nm$^{-1}$. In such a case, the 1.0 nm$^{-1}$ peak tends to be broad and the 1.4 nm$^{-1}$ peak is weaker (typically, Fig 2a and 2d). The 0.5 nm$^{-1}$ peak becomes stronger when the beam penetrates 10–15 μm below the surface. Further down, it remains distinct while the peaks at 1.0 and 1.4 nm$^{-1}$ become weaker and form a single broad peak. When they are merged, a single peak at around $q$ = 1.0 nm$^{-1}$ with a tail towards higher angles is observed (Fig 2a). Generally,

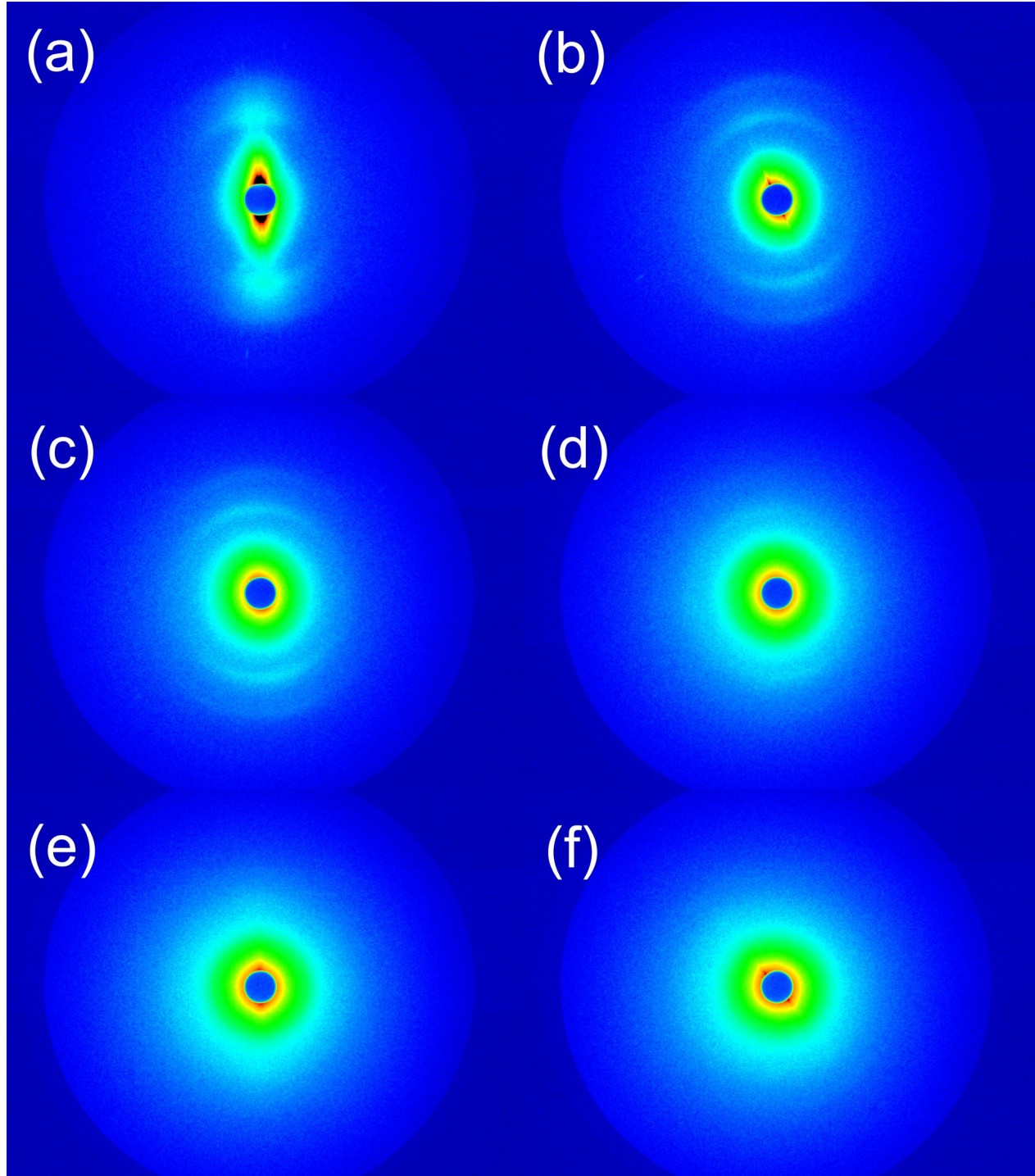

**Fig 1. X-ray diffraction patterns from human skin.** (a) A typical X-ray diffraction pattern recorded at the top of the skin where the diffraction peaks were first observed. (b)-(f) Diffraction patterns recorded when the skin was raised with 5 μm steps, moving the X-ray beam towards the bottom of SC. The skin surface is horizontal.

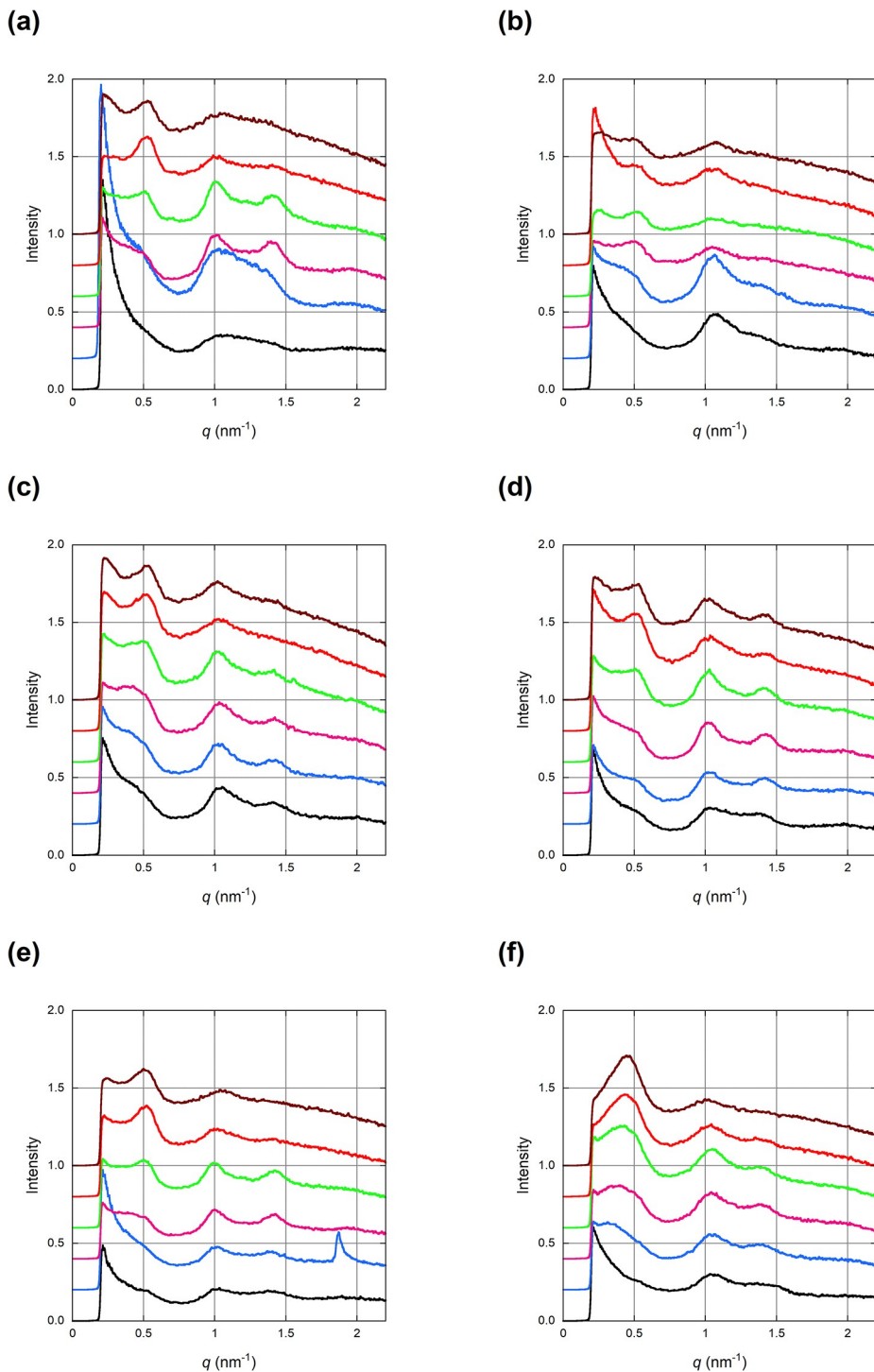

**Fig 2. Radial intensity distributions recorded at different depths of the SC.** (a)-(f) were obtained from different skin samples. The sample was raised with 5 μm steps from the black to the brown curve so that the black curve is closest to the skin surface. The profiles are vertically displaced for clarity. A peak at $q = 1.85$ nm$^{-1}$ in (e) is due to crystalline cholesterol.

the diffraction peaks from lipids were observed over the depth of 20 μm which is slightly larger than the reported thickness of SC, partly due to the size of the X-ray beam. Thus, the diffraction pattern from deeper regions of SC may include contribution from lipids newly synthesized in stratum granulosum.

The peak at $q = 1.0$ nm$^{-1}$ is slightly variable in its position across SC. When it appears together with the peak at $q = 0.5$ nm$^{-1}$, it is close to $q = 1.0$ nm$^{-1}$, while it is often around $q = 1.05$ nm$^{-1}$ when the peak at $q = 0.5$ nm$^{-1}$ is not observed in the upper part of the SC (Fig 2b and 2d). However, at the depth where the peak is highest, its position is fixed within 1% as described above.

The sixth skin sample gave an anomalous diffraction pattern. As seen Fig 2f, there is a strong and broad peak around q = 0.4 nm$^{-1}$ which moves towards higher angles at deeper regions. However, the profiles in Fig 2f are similar to others in the region q > 0.7 nm$^{-1}$. Thus, the intercellular lipid structure was the same as in other samples. The origin of the peak is unknown, but this sample may have been obtained from a person who used a particular kind of lipids for skin care which produced an additional large peak in the small angle region. This result shows variability of human skin samples and suggests danger in working only on a very small number of samples. Small variations in relative intensity of the peaks were also found among the other five samples (particularly, Fig 2b).

## Structure model

We attempted to explain the experimental results using lipid structure models. In Fig 1b, the peaks at $q = 0.5$, 1.0 and 1.4 nm$^{-1}$ appear at the medium depth of SC. The former two peaks may be indexed as first and second orders, but the peak at 1.4 nm$^{-1}$ cannot be indexed. To account for the observed diffraction profiles, models of electron density distribution were constructed. These are based on the electron micrographs of Swartzendruber et al. [13], but the distance between bands and their heights did not strictly follow the micrographs which were obtained from the sections of chemically fixed, dehydrated, and stained samples. In these electron micrographs, RuO$_4$ was used to stain lipid and thus the density mainly indicates distribution of RuO$_4$. Although RuO$_4$ is assumed to react polar lipids [19], it is uncertain which lipids are preferentially stained. The simplest unit in the model consists of a central density minimum and two minima that were at 5.0 nm on either side of the center with the same depth (Fig 3a). Profiles of all minima were assumed to be Gaussians. Diffraction intensity expected from this unit gives broad peaks at around $q = 0.55$ and 1.1 nm$^{-1}$ (Fig 3d). When two such units were superposed with a separation of 13.0 nm (Fig 3b), the resultant electron density profile resembles a model of tri-lamellar structure (the so-called Landmann unit) [13, 20]. X-ray diffraction expected from this model has peaks at $q = 0.5$, 1.0 and 1.3 nm$^{-1}$ (Fig 3d). With three units (Fig 3c), there are small additional peaks other than the major three peaks (Fig 3d). When the diffraction intensities from the one-, two- and three-unit models are added with a ratio of 3:3:1, the resultant profile (Fig 3e) resembles the observed profile with peaks at $q = 0.56$, 1.04, 1.41 nm$^{-1}$ and a broad baseline between the latter two peaks. This is similar to the experimentally observed profile in Fig 2.

Attempts were also made to use density profiles obtained by cryo-electron microscopy. Since the micrographs were obtained from unstained sections, the intensity in them should represent electron density in the specimen. However, the density distribution with five lucent bands presented in Fig 2b by Iwai et al. [8] produces peaks at $q = 0.6$ and 1.1 nm$^{-1}$ but only a very weak peak at 1.4 nm$^{-1}$. It is possible that underfocusing to obtain high contrast (Iwai, personal communication) in the micrographs emphasized the periodicity in electron density.

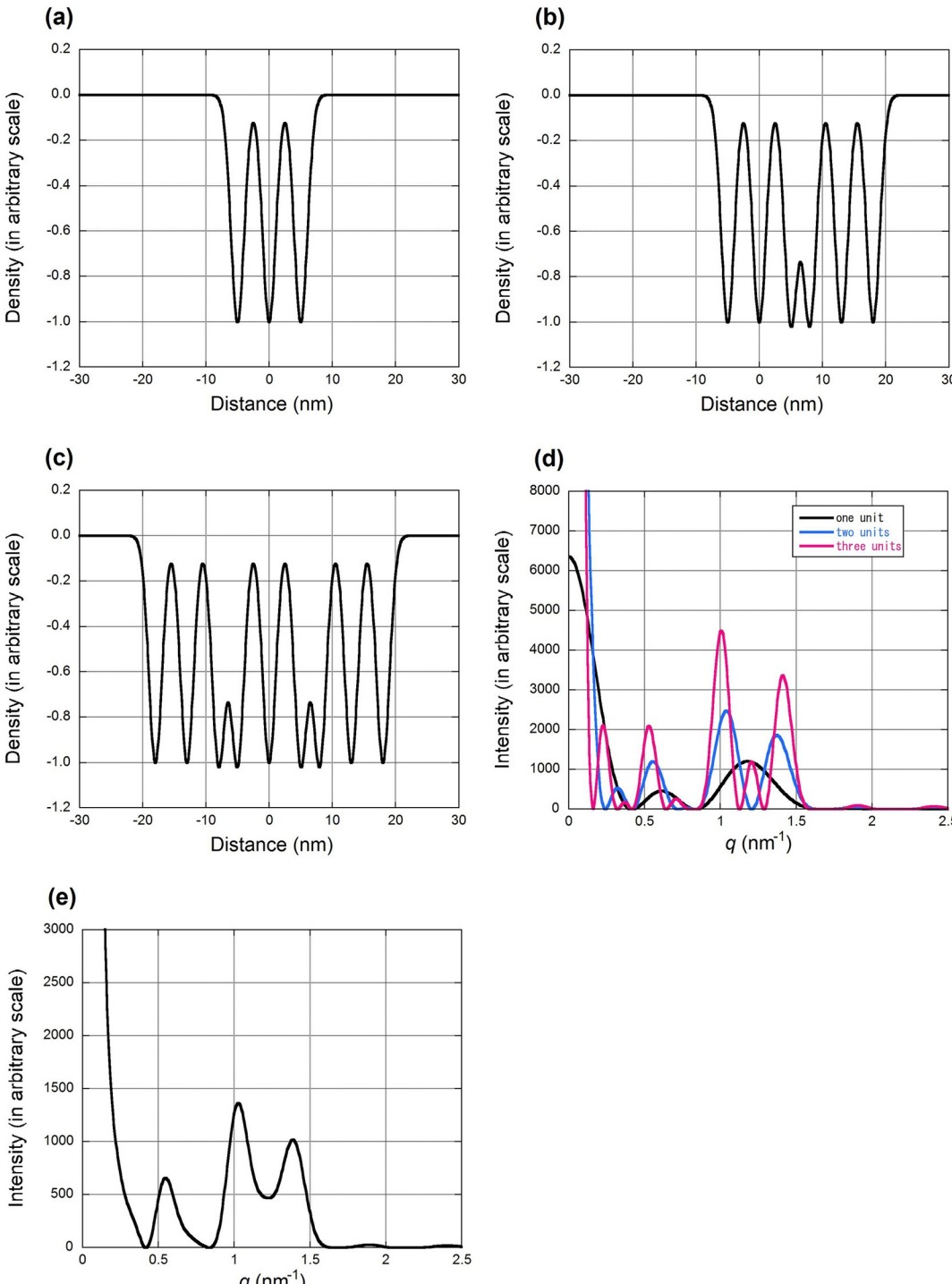

**Fig 3. Models based on electron microscopy and calculated diffraction from them.** (a) An electron density model of one lamellar unit. There is an electron sparse region, corresponding hydrocarbon chains of lipids, in the center and there are two such regions at 5 nm on each side. All low electron density regions are simulated by a Gaussian with a full-width at half maximum of 3.5 nm. (b) A model in which the model (a) was duplicated in the right with a shift of 13 nm. (c) A model with three identical units. The third unit is shifted by -13 nm from the first unit. (d) X-ray diffraction expected from the two-unit model (black) and the three-unit model (blue). (e) Mixture of one-, two-, and three-unit models with a ratio of 3:3:1.

## Discussion

In this study, we used human skin samples that had been kept wet and unfrozen until the experiment. Compared with the results of a study which used a thin section of a skin that had been kept frozen [6], the X-ray diffraction pattern obtained with the present technique gave more distinct peaks (Fig 1). Possibly, freezing may affect the lipid structure, but there may be also damages caused in the cutting procedure. As other factors that may influence the diffraction pattern, the sample setting is also slightly different, and the tension on the skin in the present experiment may have induced orientation. Similarities to the diffraction pattern from a live mouse skin [15], which was obtained with the same sample setting as in the present work, suggest that the current sample handling technique maintained the SC structure well and this lamellar structure is common in mammalian skins. Possible effects of a long period in preservation solution on the gross lamellar structure is not evident in comparison with the results on a live mouse skin.

From the crystallographic consideration described in detail in S1 Appendix, the change in the diffraction pattern observed when the beam moves deeper into the SC mainly represents differences in the lipid structures at different depths. The pattern obtained close to the skin surface represents diffraction purely from the upper part of SC. The patterns at deeper depths are summation of ones coming from different parts along the beam, but the largest contribution comes from the part where the layers of intercellular lipids lie parallel to the beam.

Although the separations between lucent bands in electron micrographs such as 11, 6.5, 4.3 nm [8, 20] sometimes coincide with the d-spacings of the diffraction peaks, such correspondence may not explain the origin of X-ray diffraction peaks, because the observed diffraction profile is a Fourier transform of the electron density distribution. Generally, distance between density maxima or minima does not directly correspond to an X-ray diffraction peak unless it is repeated regularly. Thus, we constructed structural models based on electron microscopy to account for the major features of the diffraction pattern (Fig 3).

Based on the discussion by Swartzendruber et al. [13], the unit in our model represents corneocyte lipid envelopes and their shared monolayer, while the two-unit model corresponds to the so-called Landmann unit [13, 21] that is made of two closely apposed bilayers. It has been shown that intercellular spaces are mostly filled by either one, two, or three such units. It was indeed found in this study that the X-ray diffraction pattern from SC can be explained by summation of diffraction patterns from different numbers of units.

It has been reported that there are two periodicities in lipids in SC. One is a long periodicity of 11–13 nm that gives rise to an order of diffraction peaks at multiples of $q = 0.5$ nm$^{-1}$ and a short periodicity of 6 nm that gives rise to a peak at $q = 1.0$ nm$^{-1}$ [3]. Neither of the two periodicities can account for the peak at $q = 1.4$ nm$^{-1}$ ($d = 4.5$ nm). The present simulation shows that such observation may be accounted for by a single phase of lipids made of two and three repeats of the basic unit. When the number of repeats is small, the simulation predicts three peaks (at $q = 0.5$, 1.0 and 1.4 nm$^{-1}$). Also, the broad intensity between the peaks at $q = 1.0$ and 1.4 nm$^{-1}$ may be explained by diffraction from the one-unit structure. Differences in the relative intensity of peaks observed in different samples may be explained by different ratios of the three structures. The broad intensity peak at $q = 1.0$ nm$^{-1}$ and a shift of the 0.5 nm$^{-1}$ peak towards higher angles at the lower part of SC (Fig 1b) suggest less contribution from the two- and three-unit structures. Thus, the molecular arrangements in intercellular lipids may change across SC.

Although this simple model accounts for major features of X-ray diffraction from SC, it is still imperfect. For example, the diffraction profiles at the top and bottom parts of SC are not satisfactorily explained by different ratios in summation. Lipid molecules are covalently

attached to corneocyte envelope [22], so that the electron density distributions with different numbers of the units may not be represented by simple superposition of the same density profile. However, a more detailed analysis was not attempted because there were considerable variations in the experimentally observed X-ray diffraction patterns recorded at different locations on the skin, suggesting there may be other contributions such as the intercellular lipids with a short periodicity [3] or materials other than intercellular lipids.

## Conclusions

We established a technique to study structure of SC in intact human skin with X-ray diffraction, which is convenient and can be applied to various skin samples. It can be used to study effects of environmental changes and drugs on lipid structure in skin. We demonstrated that the obtained diffraction pattern can be analyzed with a simple model based on the tri-lamellar structure.

## Supporting information

**S1 Appendix. Interpretation of the X-ray diffraction.**
(DOCX)

**S1 Fig.** (a) (top) A schematic view of SC with a "brick-and-mortar" model. The "bricks" (green) are keratynocytes and the "mortar" (orange) is intercellular lipids. (bottom) In the mortar, lipid molecules (black sticks) are arranged across the gap. (b) A flat skin sample (top) is folded (bottom) for X-ray diffraction measurement. SC (blue) is at the top of the fold, which is supported by cellular tissues (grey). The SC is investigated with an X-ray beam (red). (c) X-ray penetration at different depths of the SC. Orange arcs represent the intercellular lipid layers that run approximately parallel to the skin surface. The X-ray beam passes along the lipid layers at the tip of the sample. In the deeper region of the SC, the lipid layers are inclined towards or away from the beam in the areas where the beam enters or leaves the sample, but still parallel in the middle. (d) Principle of X-ray diffraction of a lamellar sample in reciprocal space. The two blue disks represent diffraction spots from lipid molecules arranged with a periodicity of 12.5 nm. The radius of the disk, which increases proportionally towards higher reflection orders, is determined by the size of coherent areas of lipid layers. Red lines are on the surface of the Ewald sphere whose radius is the reciprocal of the X-ray wavelength. Intersection of the sphere with the disk is drawn in red broken line. It is this intersection that gives rise to diffraction. (e) Effect of tilt of a skin sample in reciprocal space. If there were no disorder in the lipid layers (i.e. if the coherent area was infinitely large) the disks in reciprocal space would become points and there would be no intersection with the Ewald sphere, hence no diffraction would be observed. However, in reality, because of the limited size of the coherent area, the disks in reciprocal space intersect the Ewald sphere (left). When the lipid layers are too inclined towards the beam, the intersection is lost and no diffraction is observed. In the present case, since the radius of the disk corresponds to about 8 degrees, no reflection is observed when the lipid layers are tilted more than 8 degrees (right).
(DOCX)

**S2 Fig. X-ray diffraction patterns from human skin.** (a) A typical X-ray diffraction pattern from human skin at a depth of about 10 μm from the surface of the skin. The sharp spots at $q = 1.85$ nm$^{-1}$ in the lower half can be attributed to cholesterol. (b) A diffraction pattern from a more disordered human sample.
(DOCX)

## Acknowledgments

We would like to thank Prof. Ichiro Hatta, Dr. Ichiro Iwai (Shiseido Co.), Dr. Takuji Kume (Kao Co.) and Dr. Ryuji Ohgaki (Osaka University) for discussion. An initial trial experiment on mouse skin was conducted with an approval of SPring-8 Program Review Committee (2013A1097).

## Author Contributions

**Conceptualization:** Naoto Yagi.

**Data curation:** Naoto Yagi, Koki Aoyama, Noboru Ohta.

**Formal analysis:** Naoto Yagi, Noboru Ohta.

**Funding acquisition:** Naoto Yagi.

**Investigation:** Naoto Yagi.

**Project administration:** Naoto Yagi.

**Software:** Naoto Yagi, Koki Aoyama, Noboru Ohta.

**Writing – original draft:** Naoto Yagi.

**Writing – review & editing:** Naoto Yagi.

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
