## [Decision Letter · Decision Letter 0]

12 Feb 2020

PONE-D-19-36009

Microbeam X-ray Diffraction Study of Lipid Structure in Stratum Corneum of Human Skin

PLOS ONE

Dear Dr. Yagi,

Thank you for submitting your manuscript to PLOS ONE. After careful consideration, we feel that it has merit but does not fully meet PLOS ONE’s publication criteria as it currently stands. Therefore, we invite you to submit a revised version of the manuscript that addresses the points raised during the review process.

Please consider carefully the comments from the reviewers and recommendations for change, and address each of these recommendations. Where you disagree with a particular comment from the reviewers please identify this with a justification for your disagreement. 

We would appreciate receiving your revised manuscript by Mar 28 2020 11:59PM. To enhance the reproducibility of your results, we recommend that if applicable you deposit your laboratory protocols in protocols.io, where a protocol can be assigned its own identifier (DOI) such that it can be cited independently in the future. For instructions see: http://journals.plos.org/plosone/s/submission-guidelines#loc-laboratory-protocols

We look forward to receiving your revised manuscript.

Kind regards,

Richard G. Haverkamp, PhD

Academic Editor

PLOS ONE

Journal Requirements:

Reviewers' comments:

Reviewer's Responses to Questions

**Comments to the Author**

1. Is the manuscript technically sound, and do the data support the conclusions?

Reviewer #1: Yes

Reviewer #2: Partly

Reviewer #3: Partly

2. Has the statistical analysis been performed appropriately and rigorously? 

Reviewer #1: Yes

Reviewer #2: No

Reviewer #3: I Don't Know

3. Have the authors made all data underlying the findings in their manuscript fully available?

Reviewer #1: Yes

Reviewer #2: No

Reviewer #3: No

4. Is the manuscript presented in an intelligible fashion and written in standard English?

Reviewer #1: Yes

Reviewer #2: Yes

Reviewer #3: Yes

5. Review Comments to the Author

Reviewer #1: This manuscript uses x-ray diffraction on natural abdominal skin of healthy patients to probe the lipid organization of the SC. It is a nice set of experiments that can be compared to past work on mice and extracted SC lipid studies. Overall, this appears to confirm similar work by Bouwstra and others that there is an existence of a short and long periodicity phase.

However, one aspect that is unclear is how the preserving fluid might influence the structure of the SC lipid phase. Clearly this work does not suffer from hydration issues but could the preserving fluid alter the lipid phase? Some discussion on this should be provided.

Reviewer #2: The authors propose that X-ray diffraction can be used to study the structure of lipids in the stratum corneum of the skin. They present preliminary data in the current study that supports the initial assumption. Fresh skin, not previously frozen, was examined using SAXS in the area of the stratum corneum. Diffraction peaks were observed at the lipid range. The results are at preliminary level, but are quite interesting. There are a few points that have to be addressed:

1. The appendix is an extended discussion of the data findings and should be included in the Discussion section of the paper.

2. In the appendix at the bottom of the second page, the authors are attempting to explain the difference in the diffraction patterns between different spots in the tissue. It is not very clear and it is rather confusing. They use wording such “wavier” that does not really apply. The fact that the difference between having defined “lines” or diffuse “arcs” in a diffraction pattern indicates that there are highly ordered structures aligned towards one direction or different directions, respectively was not mentioned. This section has to be rewritten and corrected.

3. In the results section at line 174 the authors highlight the risk of working with a small amount of samples. However, only 6 samples were used in the current study. The authors should consider including more samples in the presented manuscript in order to back up their data.

4. In the same section as above, the authors mention that the last sample was different to the rest ones and it is assumed that the donor of this particular sample might have had a particular condition. In addition to more samples, the study should consider the age and condition of donors and create different groups for comparison.

5. The data presented in the current study is interesting. However, it should be considered as preliminary data and more effort should be invested in order to make it more complete. As it was mentioned already more samples will be necessary in order to prove that the results are consistent and not circumstantial. Electron microscopy technique should be employed at the same samples in order to back up the X-ray diffraction results, especially to show the different directions the lipids can be arranged within the layer. Microscopy is referred in the manuscript, but it was not performed in the samples that were used in the current study.

Reviewer #3: I read the manuscript Microbeam X-ray Diffraction Study of Lipid Structure in Strstum corneum of Human Skin with great interest. The manuscript reports on the diffraction patterns as a function of depth in ststum corneum of fresh skin.

Although the manuscript reports on the X-ray diffraction curves as a function of depth in stratum corneum and reveals different profiles at different depths, which is of interest, this reviewer has major comments on this manuscript. These are listed below:

Major comments

Introduction: the introduction is not very well written and does not explain what is known in the field of stratum corneum lipids.

For example, there is one publication in which isolated fresh skin has been measured and compared with isolated stratum corneum from the same donor using X-ray diffraction and showed that the diffraction profiles are very similar. (Schreiner et al, Journal Invest. Dermatology, 2000) Therefore, it is absolutely not done to suggest that trypsin digestion changes the properties of the stratum corneum. If you suggest this, please show the data, that means measure using the same donors isolated stratum corneum and fresh skin and compare the curves.

The authors cannot simply argue that hydration has been carried out under unnatural conditions. What do the authors mean by this? Specify the papers describing this, but do not make a general comment. Most studies have been done by hydrating the stratum corneum at room temperature at a fixed relative humidity. In the studies described here, probably the stratum corneum was fully hydrated (Transport in medium), which is also not natural. Especially having the skin for a longer time period at high humidity may have an effect.

Many more studies on stratum corneum have been performed (mouse skin, pig skin, human skin equivalents) sometimes also as function of temperature, in several cases showing that the peaks disappear at the same temperature, indicating that these are attributed to the same lamellar phase. Nothing is mentioned about this. Also studies using isolated pig or human ceramides are relevant as these provide also useful information.

Methods:

Although it is excellent to see the curves as function of depth, these curves have already been changed by subtracting the curve obtained at perpendicular orientation in which a minimum diffraction of the lipids has been detected. However, I would like to see the original curves for at least two reasons:

a. The lipid peaks are very broad, which makes subtraction a difficult procedure.

b. The scattering at low angle is very steep, which make subtraction also a difficult procedure.

I cannot rule out that peak positions are sensitive to this procedure.

Results

Model calculations.

The authors use the RuO4 profiles of Swarzendruber to calculate the intensities of the peaks. However, these profiles are in fact a print of the real structure as it visualizes the position of RuO4. It is even not clear to which parts of the lipids RuO4 is fixed. So no information can be drawn about the underlaying structure, only that there is a certain repeat in the structure. This should be very clearly stated.

Minor comments

Page 3: electron diffraction cannot detect the lamellar structures

If the curvature may effect the diffraction profile, why not measuring with a straight oriented sample.

If the beam location gradually changes with 5 micrometer steps, then the total length over which has been measured is 30 micrometer (including the size of the beam).

Line 136 sentence is a repeat of the previous sentence. Second part of thst sentence is not clear.

Line 146/147: The accuracy of the q values is not realistic. Later on it is explained there are differences between donors. This is probably standard deviation of the mean? Not taking into account different donors? See remark line 166

Line 221: freezing of skin samples can induce holes in the lipid structure and therefore the repeating pattern is interrupted.

Line 249: In many other publications Bouwstra always attributed the 1.4 nm-1 peak to the 12-13 nm lamellar phase. So this remark is quite biased and should be changed.

Line 261: there is no plasma membrane in strstum corneum

6. PLOS authors have the option to publish the peer review history of their article (what does this mean?). If published, this will include your full peer review and any attached files.

Reviewer #1: No

Reviewer #2: No

Reviewer #3: No

---

## [Author Response · Author response to Decision Letter 0]

9 Apr 2020

I would like to thank the three reviewers for valuable comments to improve the manuscript. It has been revised to accommodate the comments. For each comment, a response is provided below.

In the revised manuscript, a few errors and typos were corrected. Particularly, the number of scans on each sample was in fact between five and ten. Previously it was described as 10 to 20, but this includes scans that were made to confirm radiation damage or test scanning software.

All the changed parts are in red.

The line numbers refer to those in the revised manuscript.

Reviewer #1: This manuscript uses x-ray diffraction on natural abdominal skin of healthy patients to probe the lipid organization of the SC. It is a nice set of experiments that can be compared to past work on mice and extracted SC lipid studies. Overall, this appears to confirm similar work by Bouwstra and others that there is an existence of a short and long periodicity phase. 

However, one aspect that is unclear is how the preserving fluid might influence the structure of the SC lipid phase. Clearly this work does not suffer from hydration issues but could the preserving fluid alter the lipid phase? Some discussion on this should be provided. 

The samples were investigated eight days after surgery [line 88]. Storage in a preservation solution might alter the lipid lamellar structure, but there is no way to study the effect. One point we would like to make is that the X-ray diffraction profile obtained in the present study is similar to that obtained in a previous study on skin of live wildtype mouse (Nakagawa et al., 2012) [lines 233-235]. 

Reviewer #2: The authors propose that X-ray diffraction can be used to study the structure of lipids in the stratum corneum of the skin. They present preliminary data in the current study that supports the initial assumption. Fresh skin, not previously frozen, was examined using SAXS in the area of the stratum corneum. Diffraction peaks were observed at the lipid range. The results are at preliminary level, but are quite interesting. There are a few points that have to be addressed: 

1. The appendix is an extended discussion of the data findings and should be included in the Discussion section of the paper. 

The discussion was summarized in Results section in the previous manuscript. As it is hard to move the entire discussion in the appendix to the manuscript, this summary is now moved to Discussion [lines 236-242]. A brief mention to Appendix is added to Data analysis section [line 107].

2. In the appendix at the bottom of the second page, the authors are attempting to explain the difference in the diffraction patterns between different spots in the tissue. It is not very clear and it is rather confusing. They use wording such “wavier” that does not really apply. The fact that the difference between having defined “lines” or diffuse “arcs” in a diffraction pattern indicates that there are highly ordered structures aligned towards one direction or different directions, respectively was not mentioned. This section has to be rewritten and corrected. 

Thank you for the suggestion. Difference between lines and arcs is now explained and this section was rewritten.

3. In the results section at line 174 the authors highlight the risk of working with a small amount of samples. However, only 6 samples were used in the current study. The authors should consider including more samples in the presented manuscript in order to back up their data. 

I understand your concern, but the samples used in this study were obtained in cosmetic surgery to remove fat in obese women. Thus, they are all from Caucasian women in their 30s to 50s. It is hard to expand this study because skin samples from other parts of a body or other sex, age or ethnic groups of donors are hard to obtain. This situation is now explained at line 90. Getting skin samples from volunteer donors involves many ethical regulations which are very tight these days, although the situation may be different in each country. The present study was made on limited samples, but it still provides a general view on lipid structure in human skin. 

4. In the same section as above, the authors mention that the last sample was different to the rest ones and it is assumed that the donor of this particular sample might have had a particular condition. In addition to more samples, the study should consider the age and condition of donors and create different groups for comparison. 

As explained above, it is difficult to obtain samples from different age groups. However, we do not think the intercellular lipid structure in the last sample was different from others. The previous manuscript only described the difference. An additional description on similarity is now added at lines 172-173.

5. The data presented in the current study is interesting. However, it should be considered as preliminary data and more effort should be invested in order to make it more complete. As it was mentioned already more samples will be necessary in order to prove that the results are consistent and not circumstantial. Electron microscopy technique should be employed at the same samples in order to back up the X-ray diffraction results, especially to show the different directions the lipids can be arranged within the layer. Microscopy is referred in the manuscript, but it was not performed in the samples that were used in the current study. 

The lipid lamellar structure discussed in this paper is a very basic one and generally considered invariable among humans, even among mammalians to some extent [lines 43-44]. There are small differences among different people as can be noticed in Fig 2, but the basic lipid structure shown in Fig 3 can be used to explain diffraction from humans as well as mammalians. Other techniques also support this view. For example, Swartzendruber et al. found common lamellar structures in mouse, pig, and human skins and proposed a molecular model that was used to construct the model in the present study [line 187]. Detailed structures may be different because lipid compositions are variable among different mammalian species. This needs further studies.

Reviewer #3: I read the manuscript Microbeam X-ray Diffraction Study of Lipid Structure in Strstum corneum of Human Skin with great interest. The manuscript reports on the diffraction patterns as a function of depth in ststum corneum of fresh skin. 

Although the manuscript reports on the X-ray diffraction curves as a function of depth in stratum corneum and reveals different profiles at different depths, which is of interest, this reviewer has major comments on this manuscript. These are listed below: Major comments 

Introduction: the introduction is not very well written and does not explain what is known in the field of stratum corneum lipids. 

For example, there is one publication in which isolated fresh skin has been measured and compared with isolated stratum corneum from the same donor using X-ray diffraction and showed that the diffraction profiles are very similar. (Schreiner et al, Journal Invest. Dermatology, 2000) Therefore, it is absolutely not done to suggest that trypsin digestion changes the properties of the stratum corneum. If you suggest this, please show the data, that means measure using the same donors isolated stratum corneum and fresh skin and compare the curves. 

Thank you for pointing out this important paper. I agree there is no evidence that trypsin treatment changes the lipid structure. Thus, the statement is retracted. The paper is now cited as a previous study on human skin [line 43, line 153]

The authors cannot simply argue that hydration has been carried out under unnatural conditions. What do the authors mean by this? Specify the papers describing this, but do not make a general comment. Most studies have been done by hydrating the stratum corneum at room temperature at a fixed relative humidity. In the studies described here, probably the stratum corneum was fully hydrated (Transport in medium), which is also not natural. Especially having the skin for a longer time period at high humidity may have an effect. 

Introduction was revised according to the comments [lines 42-45]. A possible problem related to transport in preservation solution is mentioned and discussed [lines 233-235].

Many more studies on stratum corneum have been performed (mouse skin, pig skin, human skin equivalents) sometimes also as function of temperature, in several cases showing that the peaks disappear at the same temperature, indicating that these are attributed to the same lamellar phase. Nothing is mentioned about this. Also studies using isolated pig or human ceramides are relevant as these provide also useful information. 

These points are mentioned in Introduction [lines 40-46].

Methods: 

Although it is excellent to see the curves as function of depth, these curves have already been changed by subtracting the curve obtained at perpendicular orientation in which a minimum diffraction of the lipids has been detected. However, I would like to see the original curves for at least two reasons: 

a. The lipid peaks are very broad, which makes subtraction a difficult procedure. 

b. The scattering at low angle is very steep, which make subtraction also a difficult procedure. 

 I cannot rule out that peak positions are sensitive to this procedure. 

In the data analysis presented in this paper, background subtraction was not performed. This is corrected now. The diffraction profiles presented in this paper may look rather flat. This is because the profiles were obtained by circular summation (not average) and because a Lorenz factor was applied. This is the reason the higher angle region tends to be enhanced. This procedure is explained [lines 108-112].

Results 

 Model calculations. 

The authors use the RuO4 profiles of Swarzendruber to calculate the intensities of the peaks. However, these profiles are in fact a print of the real structure as it visualizes the position of RuO4. It is even not clear to which parts of the lipids RuO4 is fixed. So no information can be drawn about the underlaying structure, only that there is a certain repeat in the structure. This should be very clearly stated. 

This point is now explained [lines 189-192].

Minor comments 

Page 3: electron diffraction cannot detect the lamellar structures 

This is corrected [line 38].

If the curvature may effect the diffraction profile, why not measuring with a straight oriented sample. 

It is hard to obtain a flat, oriented skin, as explained in Appendix page 2-3.

If the beam location gradually changes with 5 micrometer steps, then the total length over which has been measured is 30 micrometer (including the size of the beam). 

This is now written in experimental [lines 122-123].

Line 136 sentence is a repeat of the previous sentence. Second part of thst sentence is not clear. 

This is a repetition. It is removed.

Line 146/147: The accuracy of the q values is not realistic. Later on it is explained there are differences between donors. This is probably standard deviation of the mean? Not taking into account different donors? See remark line 166 

These are in fact standard deviations. It is rather surprising the SD is as small as 1%. The range of values is now given (0.504-0.515 for q=0.511, and 1.055-1.019 for q=1.034). The variation in peak position described at lines 165-167 took place when different depth of stratum corneum was interrogated. At the depth where the peaks are strongest, their positions were not variable [lines 168-169].

Line 221: freezing of skin samples can induce holes in the lipid structure and therefore the repeating pattern is interrupted. 

This possibility is pointed out [line 226].

Line 249: In many other publications Bouwstra always attributed the 1.4 nm-1 peak to the 12-13 nm lamellar phase. So this remark is quite biased and should be changed. 

This statement was removed.

Line 261: there is no plasma membrane in strstum corneum

This is corrected [line 275].

---

## [Editor Report · Decision Letter 1]

29 Apr 2020

Microbeam X-ray Diffraction Study of Lipid Structure in Stratum Corneum of Human Skin

PONE-D-19-36009R1

Dear Dr. Yagi,

We are pleased to inform you that your manuscript has been judged scientifically suitable for publication and will be formally accepted for publication once it complies with all outstanding technical requirements.

With kind regards,

Richard G. Haverkamp, PhD

Academic Editor

PLOS ONE
---

## [Editor Report · Acceptance letter]

1 May 2020

PONE-D-19-36009R1 

Microbeam X-ray Diffraction Study of Lipid Structure in Stratum Corneum of Human Skin 

Dear Dr. Yagi:

I am pleased to inform you that your manuscript has been deemed suitable for publication in PLOS ONE. Congratulations! Your manuscript is now with our production department. 

With kind regards,

on behalf of

Professor Richard G. Haverkamp 

Academic Editor

PLOS ONE